# Colon Bioaccessibility under In Vitro Gastrointestinal Digestion of a Red Cabbage Extract Chemically Profiled through UHPLC-Q-Orbitrap HRMS

**DOI:** 10.3390/antiox9100955

**Published:** 2020-10-06

**Authors:** Luana Izzo, Yelko Rodríguez-Carrasco, Severina Pacifico, Luigi Castaldo, Alfonso Narváez, Alberto Ritieni

**Affiliations:** 1Department of Pharmacy, University of Naples “Federico II”, Via Domenico Montesano 49, 80131 Naples, Italy; luigi.castaldo2@unina.it (L.C.); alfonso.narvaezsimon@unina.it (A.N.); alberto.ritieni@unina.it (A.R.); 2Laboratory of Food Chemistry and Toxicology, Faculty of Pharmacy, University of Valencia, Av. Vicent Andrés Estellés s/n, 46100 Burjassot, Valencia, Spain; yelko.rodriguez@uv.es; 3Department of Environmental, Biological and Pharmaceutical Sciences and Technologies, University of Campania “Luigi Vanvitelli”, Via Vivaldi 43, 81100 Caserta, Italy; severina.pacifico@unicampania.it; 4Health Education and Sustainable Development, Federico II University, 80131 Naples, Italy

**Keywords:** red cabbage, in vitro gastrointestinal digestion, antioxidants, acid-resistant capsule, bioaccessibility, UHPLC-Q-Orbitrap HRMS

## Abstract

Red cabbage is a native vegetable of the Mediterranean region that represents one of the major sources of anthocyanins. The aim of this research is to evaluate the antioxidant capability and total polyphenol content (TPC) of a red cabbage extract and to compare acquired data with those from the same extract encapsulated in an acid-resistant capsule. The extract, which was qualitatively and quantitatively profiled by UHPLC-Q-Orbitrap HRMS analysis, contained a high content of anthocyanins and phenolic acids, whereas non-anthocyanin flavonoids were the less abundant compounds. An in vitro gastrointestinal digestion system was utilized to follow the extract’s metabolism in humans and to evaluate its colon bioaccessibility. Data obtained showed that during gastrointestinal digestion, the total polyphenol content of the extract digested in the acid-resistant capsule in the Pronase E stage resulted in a higher concentration value compared to the extract digested without the capsule. Reasonably, these results could be attributed to the metabolization process by human colonic microflora and to the genesis of metabolites with greater bioactivity and more beneficial effects. The use of red cabbage extract encapsulated in an acid-resistant capsule could improve the polyphenols’ bioaccessibility and be proposed as a red cabbage-based nutraceutical formulation for counteracting stress oxidative diseases.

## 1. Introduction

Red cabbage (*Purple Brassica oleracea* L. var. *capitata F. rubra*) is a native vegetable of the Mediterranean region that originated in Europe in the 16th century and nowadays is largely consumed worldwide. Among different vegetables in the human diet, it represents one of the major sources of polyphenolic compounds, especially anthocyanins [1]. Anthocyanins are natural glycoside compounds belonging to the flavonoids group, and they are mainly responsible for the colors of fruits, vegetables, and flowers. The flavylium (2-phenylchromenylium) ion represents the basic molecular skeleton of anthocyanins, whose sugar-free components are anthocyanidin aglycones. Among naturally anthocyanidins known until now, merely six principal types are common in fruit and vegetables, which mainly differ in the oxygenation degree of the flavonoid B-ring (e.g., hydroxyl and/or methoxyl groups). Thus, anthocyanins variability in plants is mainly due to the sugar moieties identity and number as well as to the diversity in acylated substituents [both aromatic (largely hydroxycinnamic acids, and/or simple C_6_C_1_ acids) or aliphatic acids (e.g., malonic acid)], which could be linked to the anthocyanin core or directly to the anthocyanidin nucleus. Particularly, the red cabbage matrix contains cyanidin 3-diglucoside-5-glucoside derivatives highly conjugated with sugars such as glucose and xylose and acyl groups including caffeoyl, *p*-coumaroyl, feruloyl, *p*-hydroxybenzoyl, sinapoyl, and oxaloyl [2,3].

The overproduction of reactive oxygen and nitrogen species (ROS/RNS) could occur in living organisms at an uncontrolled rate, defining oxidative stress condition onset, which is correlated with various forms of health damage including chronic age-related diseases, atherosclerosis, carcinogenesis, and neurodegenerative disorders. A healthy nutrition, mainly based on fruits and vegetables, is suggested to delay or positively modulate the dynamic balance between oxidants and antioxidants, thanks to plant foods diversity in bioactive compounds, such as polyphenols. Polyphenols are exogenous antioxidant compounds that are able to prevent and/or inhibit the genesis of pathophysiological perturbations in redox circuitry. According to Sies et al. [4], the oxidative stress represents “a disturbance in the pro-oxidant–antioxidant balance in favor of the former, leading to potential damage”. Anthocyanins are known to have a wide range of health-promoting properties for human health including cytoprotective activity, which might be due to the ability of anthocyanins to decrease cell death, lactate dehydrogenase (LDH) release, caspase 3 activation, and DNA damages [5]. Moreover, scientific reports support an increase in the cytoprotective effect as a result of the anthocyanin-rich diet [6].

To fully preserve or augment polyphenols bioactivity and achieve an efficient therapeutic activity, these compounds often need to be formulated into bioavailable dosage nutraceutical forms [7,8]. Indeed, new formulations by using polyphenols from dietary plants are continuously investigated also for safeguarding polyphenol chemical features that could be compromised during their metabolism fates in humans. This is particularly true for anthocyanins and anthocyanidin compounds, which are highly instable and very sensitive to degradation by oxygen, temperature, light, enzymes, and pH, and whose availability after gastrointestinal digestion could limit their beneficial effect on health.

The correlation of the anthocyanins’ consumption and beneficial effects on human health has been reported by many scientific studies and includes sundry protective effects on human health such as antioxidant, anti-inflammatory, anticancer activity, and effects on the cardiovascular, neurodegenerative, and metabolic systems [9,10,11]. To confer beneficial health effects, bioactive compounds need to be bioavailable and reach, after gastrointestinal (GI) digestion, target tissues in the human body. Although fruits and vegetables are an abundant source of polyphenols and other bioactive compounds, the available amount of these substances after small intestine digestion is significantly reduced [12].

Anthocyanins are stable in the acid conditions existing in the stomach, whereas their stability decreases with the increase in the pH value in the small intestine. Therefore, the bioavailability of these compounds is affected [13]. In particular, it was observed that at pH > 7, anthocyanidins undergo degradation, and that the presence of sugar moieties in anthocyanins favors an increase in terms of stability at neutral pH in respect to their aglycones [14]. Thus, diglycosides are more stable than monoglycosides, which is explained by the hindrance from sugar parts that prevent the degradation into phenolic acids and aldehydes [2]. Accordingly, it is important to define the effect of such polyphenols and their stability during the digestion process and consequentially, their bioaccessibility and their possible beneficial effects. Until now, limited information describing the in vivo effects of the GI process on dietary polyphenols has been reported [15,16,17].

Thus, the aim of this scientific research is as follows: (i) to prepare red-cabbage extracts based on different extracting mixtures, evaluated for their total polyphenol content (TPC) and the antiradical and reducing properties; (ii) to compare the antioxidant activity of red cabbage extract showing the highest TPC value with data from the same extract encapsulated in an acid-resistant capsule, and (iii) to assess the bioaccessibility of the extract, such as it is, and its encapsulated formula through an in vitro gastrointestinal digestion, with a view to propose the use of the red-cabbage nutraceutical formulation in slowing down or delaying oxidative stress onset.

The polyphenolic profile of the prepared red cabbage extract was ascertained through ultra-high-performance liquid chromatography coupled to a high-resolution Orbitrap mass spectrometry.

## 2. Materials and Methods

### 2.1. Reagents and Materials

Methanol (MeOH), ethanol (EtOH), acetic acid (AcOH), formic acid (FA), and acetonitrile (AcN) HPLC grade were purchased from Merck (Darmstadt, Germany). Deionized water (<18 MΩ x cm resistivity) was obtained from a Milli-Q water purification system (Millipore, Bedford, MA, USA).

Potassium chloride (KCl), potassium thiocyanate (KSCN), monosodium phosphate (NaH_2_PO_4_), sodium sulfate (Na_2_SO_4_), potassium persulphate (K_2_S_2_O_8_), sodium chloride (NaCl), sodium bicarbonate (NaHCO_3_), sodium carbonate (Na_2_CO_3_), hydrochloric acid (HCl), pepsin (250 U/mg solid) from porcine gastric mucosa, pancreatin (4 USP) from porcine pancreas, protease from *Streptomyces griseus*, also named Pronase E (3.5 U/mg solid), and Viscozyme L were purchased from Sigma Aldrich (Milan, Italy).

The compound of 2,2-azinobis (3-ethylbenzothiazoline-6-sulphonic acid) diammonium salt (ABTS), ferrous chloride (FeCl_2_), 1,1-Diphenyl-2-picrylhydrazyl (DPPH), 2,4,6-tris(2-pyridyl)-1,3,5-triazine (TPTZ), Folin–Ciocalteu reagent, (±)-6-Hydroxy-2,5,7,8-tetramethylchromane-2-carboxylic acid commonly called Trolox (C_14_H_18_O_4_), and gallic acid (C_7_H_6_O_5_) were acquired from Sigma Aldrich (Milan, Italy). All other chemicals and reagents were of analytical grade.

### 2.2. Sampling

Red cabbage (*Brassica oleracea L*. var. *capitata F. rubra*) plants were grown in different fields located in Campania, South of Italy. All bulbs (*n* = 10) were harvested in September 2019. After the samples arrived in the laboratory, the cabbage samples with mechanical damage and visible spoilage were separated. Red cabbage samples were rapidly washed under running tap water and chopped into small pieces before being frozen and freeze-dried (Lyovapor™ L-200, Buchi srl, Milan, Italy). After lyophilization, the dry weight of the samples obtained was recorded; then, they were milled into powder (particle size 200 µm) using a laboratory mill. The powders were stored at −80 °C until analysis. All the analyses were performed in triplicate, the replicates were independents and results expressed as mean ± SD. Dry matter content of red cabbage corresponded to 12%.

### 2.3. Red Cabbage Polyphenolic Extraction

Polyphenols were extracted according to the procedure reported by Grace et al. [18] with some modifications. Briefly, 2.5 g of freeze-dried red cabbage was introduced into a 50-mL Falcon tube (Conical Polypropylene Centrifuge Tube; Thermo Fisher Scientific, Milan, Italy) and extracted with 30 mL of five different mixtures: (1) MeOH:H_2_O (60:40) 0.1% FA; (2) MeOH:H_2_O (70:30) 0.1% AcOH; (3) MeOH:H_2_O (80:20) 0.1% FA; (4) H_2_O 0.1% FA; (5) EtOH:H_2_O (70:30) 0.1% AcOH. The samples were vortexed (ZX3; VEPL Scientific, Usmate, Italy) for 2 min and sonicated (LBS 1; Zetalab srl, Padua, Italy) for 30 min (vortexed at 10-min intervals). Then, the mixture was centrifuged for 10 min at 4000 rpm at 20 °C. The supernatant was collected and filtrated through a 0.22 µm filter. A portion of the extracts was kept in refrigeration conditions until further analysis.

Moreover, another part of the red cabbage polyphenol extract obtained with mixture 1 after lyophilization was employed for the capsules’ formulation. In particular, the capsules contained 1000 mg of red cabbage polyphenolic extract. The capsules used were acid-resistant (hydroxypropyl cellulose E464, gellan gum E418, titanium dioxide E171).

### 2.4. Determination of Total Phenolic Content (TPC)

The Folin–Ciocalteu method was used for determining the total phenolic content in accordance with the procedure reported by Izzo et al. [19]. Briefly, 500 μL of deionized water and 125 μL of the Folin–Ciocalteu reagent 2 N were added to 125 μL of red cabbage extract. The tube was mixed and incubated for 6 min in dark conditions. Then, 1.25 mL of 7.5% of sodium carbonate solution and 1 mL of deionized water were added. The reaction mixture was maintained in dark for 90 min. Finally, the absorbance at 760 nm was measured through a spectrophotometer system. Results were expressed as mg of gallic acid equivalents (GAE)/g of dry weight sample.

### 2.5. UHPLC-Q-Orbitrap HRMS Analysis

The qualitative and quantitative profile of bioactive compounds were performed by Ultra High-Pressure Liquid Chromatograph (UHPLC, Dionex UltiMate 3000, Thermo Fisher Scientific, Waltham, MA, USA) equipped with a degassing system, a Quaternary UHPLC pump working at 1250 bar, and an autosampler device. Chromatographic separation was carried out with a thermostated (T = 25 °C) Kinetex 1.7 µm F5 (50 × 2.1 mm, Phenomenex, Torrance, CA, USA) column. The mobile phase consisted of 0.1% FA in water (A) and 0.1% FA in methanol (B). The injection volume was 1 µL. The gradient elution program was as follows: an initial 0% B, increased to 40% B in 1 min, to 80% B in 1 min, and to 100% B in 3 min. The gradient was held for 4 min at 100% B, reduced to 0% B in 2 min, and another 2 min for column re-equilibration at 0%. The total run time was 13 min, and the flow rate was 0.5 mL/min.

The mass spectrometer was operated in both negative and positive ion mode by setting 2 scan events: full ion MS and all ion fragmentation (AIF). The following settings were used in full MS mode: resolution power of 70,000 Full Width at Half Maximum (FWHM) (defined for *m/z* 200), scan range 80–1200 *m/z*, automatic gain control (AGC) target 1 × 10^6^, injection time set to 200 ms and scan rate set at 2 scan/s. The ion source parameters were as follows: spray voltage 3.5 kV; capillary temperature 320 °C; S-lens RF level 60, sheath gas pressure 18, auxiliary gas 3, and auxiliary gas heater temperature 350 °C.

For the scan event of AIF, the parameters in the positive and negative mode were set as follows: mass resolving power = 17,500 FWHM; maximum injection time = 200 ms; scan time = 0.10 s; ACG target = 1 × 10^5^; scan range = 80–120 *m/z*; isolation window to 5.0 *m/z*; and retention time to 30 s. The collision energy was varied in the range of 10 to 60 eV to obtain representative product ion spectra.

For accurate mass measurement, identification and confirmation were performed at a mass tolerance of 5 ppm for the molecular ion and for both fragments. Data analysis and processing were performed using Xcalibur software, v. 3.1.66.10 (Xcalibur, Thermo Fisher Scientific, Waltham, MA, USA).

### 2.6. Antioxidant Activity

#### 2.6.1. ABTS Radical Cation Scavenging Assay

Free radical-scavenging activity was measured by using the method reported by Luz et al., [20]. Briefly, 9.6 mg of 2,2-azinobis (3-ethylbenzothiazoline-6-sulphonic acid) diammonium salt was solubilized in 2.5 mL of deionized water (7 mM) and 44 µL of solution of potassium persulfate (K_2_S_2_O_8_; 2.45 mM) were added. The solution was kept in dark conditions at room temperature for 16 h prior to use. Afterward, ABTS^•+^ solution was diluted with ethanol to reach an absorbance value of 0.70 (±0.02) at 734 nm. Then, to 1 mL of ABTS^•+^ solution with an absorbance of 0.700 ± 0.050, 0.1 mL of opportunely diluted sample was added. After 2.5 min wait, the absorbance was immediately measured at 734 nm. Results were expressed as millimoles of Trolox Equivalents (TE)/kg of dry weight sample.

#### 2.6.2. DPPH Free Radical-Scavenging Assay

The total free radical-scavenging activity of the analyzed samples was determined using the method suggested by Brand-Williams et al. [21] with modifications. Briefly, 1,1-diphenyl-2-picrylhydrazyl (4.0 mg) was solubilized in 10 mL of methanol and then diluted to reach a value of 0.90 (±0.02) at 517 nm. This solution was used to perform the assay, and 200 μL of sample extract was added to 1 mL of working solution. Results were expressed as mmol Trolox Equivalents (TE)/kg of dry weight sample.

#### 2.6.3. Ferric Reducing Antioxidant Power

The antioxidant capacity of red cabbage samples was estimated spectrophotometrically following the procedure of Benzie and Strain [22]. The ferric reducing/antioxidant power (FRAP) reagent was prepared by mixing acetate buffer (0.3 M; pH 3.6), TPTZ solution (10 mM), and ferric chloride solution (20 mM) in the proportion of 10:1:1 (*v/v/v*). Freshly prepared working FRAP reagent was used to perform the assay. Briefly, 0.150 mL of the appropriately diluted sample was added to 2850 mL of FRAP reagent. The value of absorbance was recorded after 4 min at 593 nm.

The method is based on the reduction of Fe^3+^ TPTZ complex (colorless complex) to Fe^2+^-tripyridyltriazine (intense blue color complex) formed by the action of electron-donating antioxidants at low pH. Results were expressed as mmol Trolox Equivalents (TE)/kg of dry weight sample. All the determinations were performed in triplicate.

### 2.7. In Vitro Simulated Gastrointestinal Digestion

The in vitro gastrointestinal digestion, composed by oral, gastric, and intestinal phases (both duodenal and colon phases), was performed according to the standardized in vitro digestion model (INFOGEST method) [23] with some modifications (Figure 1). The simulated salivary, gastric, and intestinal fluid was prepared in accordance with the proportion salts reported by Castaldo et al. in Table 8 of the materials and methods section.

In the oral phase, 1 g of extract and one capsule contained 1 g of extract were mixed with 3.5 mL of warmed simulated salivary fluid (SSF). Then, 0.5 mL of α-amylase enzyme (50 mg of 250 U/mg solid), 25 µL of 0.3 M CaCl_2_ (H_2_O)_2_, and 975 µL of water were added and thoroughly mixed. Afterward, the pH of the mixture was adjusted to 7 with HCl 1 M, and the sample was incubated for 2 min at 37 °C at 150 rpm in an orbital shaker (KS130 Basic IKA, Argo Lab, Milan, Italy).

In order to simulate the gastric phase, 7.5 mL of simulated gastric fluid (SGF), 1.6 mL of pepsin (59.2 mg of 4 USP), 5 µL of 0.3 M CaCl_2_ (H_2_O)_2_, and 695 µL of water were added and thoroughly mixed. The pH value was adjusted to 3 using HCl 6 M. The sample was incubated for 120 min at 37 °C at 150 rpm in an orbital shaker (KS130 Basic IKA, Argo Lab, Milan, Italy).

In the intestinal phase, 11 mL of simulated intestinal fluid (SIF), 5 mL of pancreatin (20 mg of 4 USP), 2.5 mL of bile salts (150 mg), 40 µL of 0.3 M CaCl_2_ (H_2_O)_2_, and 1300 µL of water were added and thoroughly mixed. The pH value was adjusted to 7 using NaOH 6 M. The sample was incubated for 120 min at 37 °C at 150 rpm in an orbital shaker (KS130 Basic IKA, Argo Lab, Milan, Italy).

To the end of the intestinal phase, the tube was centrifuged at 5000 rpm for 10 min. To simulate the colon digestion process, the supernatant was collected, while 5 mL of Pronase E (1 mg/mL water solution) was added to the pellet. In this step, the pH value was adjusted to 8 using NaOH 1M and incubated for 60 min at 37 °C. Afterward, the supernatant was collected, lyophilized, and stored. The residue pellet was mixed with 150 µL of Viscozyme L, and the pH value was adjusted to 4 and incubated for 16 h at 37 °C. Finally, the supernatant was collected, stored, and lyophilized whereas the pellet was eliminated. All the supernatants collected during the different in vitro digestion phases were freeze-dried and then dissolved in MeOH:H_2_O (6:4, *v/v*) containing 0.1% FA for the evaluation of antioxidant activity and total polyphenols content.

### 2.8. Statistical Analysis

Statistical analysis of data was performed by two-way ANOVA analysis (SPSS 13.0) followed by the Tukey–Kramer multiple comparison test to evaluate significant differences; *p*-values ≤ 0.05 were considered as significant. All the determinations were performed in triplicate, and results were expressed as mean ± standard deviation (SD).

## 3. Results and Discussion

In this context, considering the diversity in polyphenol compounds of red cabbage, herein, the total polyphenol content and the antioxidant capability of red cabbage extracts obtained by using five different extracting mixtures were firstly evaluated. The red cabbage extract from the hydroalcoholic solution MeOH:H_2_O (6:4, *v/v*), acidified with 0.1% FA, appeared to be the most active. Thus, this extract, which was chemically profiled by UHPLC-Q-Orbitrap HRMS, underwent an in vitro gastrointestinal digestion simulation, together with its encapsulated form. Accordingly, the effects of the GI process on the extract as it is and that formulated in an acid-resistant capsule were compared.

### 3.1. Red Cabbage Extract Bioactivity

In the first approach, efficient extraction mixtures to maximize polyphenol recovery, mainly in its anthocyanin component, were explored. The choice of the solvent represents an important step in the extraction process because of its impact on the yield of bioactive compounds and consequently on human system effects. As reported by Lapornik et al. [24], who studied the solvent effect on the extraction of anthocyanins and other polyphenols from grape and red currant, ethanol and methanol extracts resulted in a major amount of bioactive compounds than water extracts. Methanol exhibits slightly better characteristics than ethanol as an extracting solvent. Since methanol and ethanol are less polar than water, they are more effective in degrading cell walls (due to their apolar feature), favoring anthocyanins and polyphenols releasing from cells. However, it must be considered that ethanol is more suitable than methanol for a safe application in the food sector [25,26]. Taking into account these previous observations, in the current scientific research, total polyphenol content (TPC) and antioxidant activity of red cabbage extracts obtained by using five different mixtures of extraction solvents are evaluated and reported in Table 1. TPC data ranged from 15.798 to 19.986 mg GAE/g. Acidified water is less suitable for the extraction of phenolic compounds from red cabbage, followed by acidified ethanol/water mixture, which showed a 2.95% increase in extraction efficiency in respect to water extract. Hydroalcoholic solution based on methanol as an alcoholic component also differed in their TPC content, and the MeOH:H_2_O (6:4) solution acidified with 0.1% formic acid appeared to elicit the best extracting properties. Indeed, beyond the ratio of alcohol to water, the type of acid component also could affect extraction overall. In fact, considering a variation of the MeOH/water ratio from 7:3 (*v/v*) to 8:2 (*v/v*) leads to a comparable TPC yield where acetic acid is used instead of formic acid. Moreover, the phenol recovery was estimated, taking account of the relative TPC value, and it showed a percentage increase of 26.5% in MeOH:H_2_O (6:4) plus 0.1% FA with respect to acidified water extract. A similar trend was observed also assessing antioxidant activity through ABTS and DPPH antiradical tests, as well as by the ferric reducing/antioxidant power assay. Data acquired are in the range of 45.128–50.849, 23.498–36.242, and 67.759–87.095 mmol Trolox^®^/kg for ABTS, DPPH, and FRAP, respectively. It appears clear that the potential antioxidant capacity of red cabbage extract could be affected by the typology and polarity of the extraction mixture used or methodology applied [26], and a great variability could be verified when the total phenolics content from previous scientific studies in the literature was consulted.

Antioxidant activity data measured through the three spectrophotometrical assays (ABTS, DPPH, and FRAP test) are in line with those from other studies [27,28]. The Relative Antioxidant Capacity Index (RACI) was determined in accordance with the method previously reported by Pacifico et al. [29]. The standard score was calculated as the average of the standard scores obtained from the raw data for the various antioxidant methods. The TPC value highlights that MeOH:H_2_O (6:4, *v/v*) 0.1% FA extract was more active than the others (Figure 2) and that an increase in methanol amount corresponds to a gradual decrease in antioxidant capacity. As shown in Table 2, a wide variability in the TPC of different red cabbage extracts was found in the literature. The concentration ranged between 0.10 and 116.00 mg/kg dry weight (dw). Presumably, the mixture MeOH:H_2_O (6:4, *v/v*) results were a good compromise applicable to the extraction of red cabbage bioactive compounds. By increasing the percentage of methanol, the TPC content decreases. This effect could be due to the major solubility of anthocyanins, which are contained in high quantity in red cabbage, in a high percentage of acidified water, whereas polyphenols are more soluble in methanol.

Cruz et al. [40] reported higher TPC content in red cabbage extract (up to 116.00 mg/g) obtained with a hydromethanolic mixture (7:3, MeOH:H_2_O, *v:v*), whereas in the here analyzed samples, all extracts were obtained by different mixtures acidified with AcOH or FA.

Murador et al. [27] evaluated the effects of different home cooking techniques on kale and red cabbage and demonstrated that these procedures seemed to have no significant effect on TPC in red cabbage. On the contrary, the cooking process facilitated the extraction of bioactive compounds due to the capability to soften the vegetable tissues, ameliorating the activity. Moreover, Wiczkowski et al. [42] determined the antioxidant activity of red cabbage varieties (Langedijker Dauer 2, Kissendrup, Koda, Kalibos, and Langedijker Polona) in two diversified lengths of the vegetation period (year 2008, 2009). The result of antioxidant activity measured through ABTS assay ranged between 87.99 and 168.76 and from 101.06 to 169.46 mmol Trolox/kg dry weight (dw) for two diversified lengths of the vegetation period, respectively. The varieties of red cabbage obtained in 2009 were characterized by a higher ability to radical scavenge than that grown in 2008, indicating that the differences in antioxidant capacity of red cabbage occurred in a variety-dependent manner. Nevertheless, amongst *Brassica* vegetables, red cabbage, brussels sprouts, and broccoli possess the highest antioxidant capacity and the contribution to health improvement can be related to their capacity [43].

### 3.2. In Vitro Bioaccessibility of Red Cabbage Polyphenols

The bioaccessibility of total phenolic compounds, which was calculated using the Folin–Ciocalteu method, and antioxidant activity determined by DPPH, ABTS, and FRAP tests during the in vitro gastrointestinal digestion are presented in Table 3 and Table 4. The in vitro gastrointestinal digestion was evaluated according to the INFOGEST procedure until the duodenal stage. Subsequently, to reproduce the microbiota occurring in the colon phase, the combined action of Pronase E and Viscozyme L. was utilized [44,45].

An improvement of bioactivity was observed for the extract digested in an acid-resistant capsule after the colonic stage compared to the extract digested without capsule. In fact, the total polyphenols content of extract digested in an acid-resistant capsule in the Pronase E resulted in a significantly higher concentration than the extract digested without capsule: 4.434 and 0.124 mg GAE/g, respectively. A similar trend was also observed by evaluating the antioxidant activity (Table 4); encapsulation appeared to favor an increase in antiradical and reducing capability mainly at the Viscozyme L stage. In GI digestion, polyphenols may interact with food constituents and be subjected to further degradation or metabolization that could affect their uptake. The presence of dietary fiber in fruits and vegetables is able to influence and modulate the phytochemicals bioaccessibility [46]. In this context, Podsȩdek et al. [47] have evaluated the stability of red cabbage antioxidant compounds, principally anthocyanins, during GI, concluding that the latter are affected by cabbage composition and vegetable constituents, including dietary fiber. During intestinal digestion, anthocyanins’ stability is hard dependent on food matrices. Notwithstanding, compared to other matrices, red cabbage has demonstrated a greater stability [43].

Some studies showed that anthocyanins have low absorption and high metabolism, and their bioavailability is lower compared with other subclasses of polyphenols [48,49]. Despite the reduced bioavailability, Sodagari et al. [50] reported that up to 70% of anthocyanins derived by foods could reach the colon. Hence, a regular intake of foods rich in anthocyanins could result in beneficial effects on human health.

Chemical modifications occurring during gastrointestinal digestion including the activity of gut microbiota lead to a releasing of smaller compounds, the principal responsible for the increased antioxidant activity. Although polyphenols were poorly absorbed in the duodenum, they can exert their antioxidant activities in the lower gut, which is able to metabolize these compounds, generating metabolites with greater activity [43]. It is speculated that bioactive compounds could be metabolized by human colonic microflora, generating metabolites with greater bioactivity and more beneficial effects. Specifically, anthocyanins, forming the majority of the polyphenols of red cabbage, are metabolized by glycosidase from gut microflora through cleavage of the C-ring to produce easily absorbed phenolic acids. Glycosidase, in the ileum, supports metabolism and the absorption of glycosides [51]. Moreover, Chen et al. [52] demonstrated that mulberry’s anthocyanins were metabolized to phenolic acids such as chlorogenic, protocatechuic, caffeic, and ferulic acids by the action of intestinal microflora in a percentage of 46.17%.

Several studies have also demonstrated that polyphenols from foods could have dissimilar bioaccessibilities. Most of the polyphenols are stable in the acidic gastric environment and degraded due to the neutral conditions encountered in the small intestine, which contribute to a reduced uptake into blood [12]. The in vitro GI digestion represents a valid tool to understand the behavior of compounds and the amounts that are effectively subjected to mucosal absorption. By using oral administration, macromolecules derived by food need firstly to be digestive and reduced in small bioactive compounds, and after withstanding to the pH in the GI tract, they could reach the absorption site in the small intestine [53]. Polyphenols metabolized by the combined action of Viscozyme and Pronase lead to the release of smaller compounds, the principal being responsible for the increased antioxidant activity [45]. Foods, including fruits and vegetables, represent an important source of bioactive compounds having numerous beneficial health effects such as antioxidant capacity [54]. As proved by scientific studies, GI digestion plays an important role in the antioxidant capacity, because the availability of bioactive compounds is influenced by the digestion process [55]. It seems that digestion could modify the antioxidant properties of foods, but there are contrasting opinions [56,57]. The antioxidant activity of dietary supplements commonly ingested as a source of antioxidant polyphenols was investigated by Henning et al. [58]. In addition, in this latest case, the stability was evaluated by determining the total phenolic content by Folin–Ciocalteu assay, and the antioxidant capacity was assessed by using DPPH, FRAP, and ABTS tests. Although polyphenols provide the major antioxidant potency, their results highlight that digestion may alter antioxidant properties depending on polyphenol content.

Naturally, the behavior of different classes of molecules during digestion is no overall standard. This finding was broadly demonstrated by Chen et al. [54], who evaluated the total phenolic content (TPC) and antioxidant activity before and after the in vitro digestion of thirty-three fruits, deriving large variations in the results. A significant improvement in TPC after gastric step was observed for eight fruits, but for the other twenty-five, it resulted in an increase after the duodenal phase. The same trend was observed by the antioxidant activity performed through DPPH, ABTS, and FRAP tests; the values significantly increased for some fruits but decreased for others.

It is demonstrated that some classes of polyphenols are able to increase their concentration after the gastric phase, justifying their high sensitivity to alkaline conditions. In fact, the majority of antioxidant compounds are degraded by alkaline pH, causing a substantial loss in the activity after the intestinal stage [57].

Unfortunately, polyphenols had not demonstrated high long-term stability and are characterized by sensitivity to heat and light. Moreover, polyphenols present a poor bioavailability because of low water solubility [59]. Finally, some of these compounds possess a bitter and astringent taste that limits their use in food or in oral medications. To circumvent these drawbacks, some processes that are able to enhance polyphenol bioavailability have been reported [60,61,62]. The use of an acid-resistant capsule in the digestion process acted as a protective agent and allowed us to avoid the consequence of gastric ambient on the bioactive molecules. In particular, acid-resistant capsules protect bioactive compounds from degradation or alteration of their chemical structure caused by changes in pH or the action of digestive enzymes. The acid-resistant capsules represent a useful strategy to conduct bioactive molecules until the intestinal district, where the bioactive compounds are favorably absorbed and exert their activity.

### 3.3. Identification and Quantification of Red Cabbage Bioactive Compounds Analyzed through UHPLC-Q-Orbitrap HRMS

Bioactive compounds of red cabbage MeOH:H_2_O (6:4, *v/v*) extract were profiled through UHPLC-Q-Orbitrap HRMS. A total of 40 different polyphenolic compounds including phenolic acids, flavonoids and anthocyanins were tentatively identified by combining MS and MS/MS spectra (Appendix A). Experiments were carried out both in ESI^-^ and ESI^+^ mode. Phenolic acids and flavonoids exhibited better fragmentation patterns producing the deprotonated molecular ion [M-H]^-^, whereas anthocyanins were investigated in positive ESI mode. Unambiguous identification of some compounds is carried out by comparison to their relative reference pure standards. The quantitative determination of target analytes was carried out using calibration curves at eight concentration levels. Each calibration curve was prepared in triplicate. Regression coefficients >0.990 were obtained. The quantification of compounds that had no standard to generate a curve was based on a representative standard of the same group. This is the case of cyanidin derivatives, which are quantified by using the calibration curve of cyanidin 3,5-diglucoside.

Screening in full scan mass chromatograms enables the identification of untargeted compounds and retrospective data analysis. The confirmation of the structural characterization of untargeted analytes was based on the accurate mass measurement, elemental composition assignment, retention time, and MS/MS fragmentation. Sensitivity was assessed by the limit of detection (LOD) and limit of quantification (LOQ). LOD is defined as the minimum concentration that enables the analyte identification with a mass error below 5 ppm. LOQ is the lowest analyte concentration that allows the analyte quantitated at defined levels for imprecision and accuracy <20%.

Red cabbage has its own characteristic anthocyanin pattern that includes acylated anthocyanins, which affect the antioxidant activity. In particular, anthocyanins with cyanidin-3-diglucoside-5-glucoside core, non-acylated, mono-acylated, or diacylated with *p*-coumaric, caffeic, ferulic, and sinapic acids were found to be the predominant compounds, which was in accordance with data reported by several scientific studies [63,64,65]. Indeed, according to Charron et al. [66], the dominant form of anthocyanins in the investigated red cabbage extract was cyanidin-3-diglucoside-5-glucoside, which appeared as the first main peak of the investigated extract. The compound showed the [M]^+^ ion at *m/z* 773.21057 and the main MS^2^ fragment ions at *m/z* 611.06012 [M-Glc]^+^, 449.16882 [M-2Glc]^+^, and 287.05219 [M-3Glc]^+^ (Appendix A). It constituted 39.5% of the anthocyanin fraction, which represented 26% of the investigated extract. Three abundant sinapoyl derivatives of the previous compound were also tentatively identified (Table 5). In particular, cyanidin-3-(sin)-diglucoside-5-glucoside and cyanidin-3-(sin)sophoroside-5-glucoside shared the molecular formula C_44_H_51_O_25_ and the neutral loss of a sinapic acid-H_2_O moiety, whereas cyanidin-3(sin)(sin)sophoroside-5-glucoside showed the [M]^+^ ion at *m/z* 1185.32959 (Appendix A). Cyanidin-3-(caf)(sin)soph-5-glucoside showed the [M]^+^ ion at *m/z* 1141.30261, and cyanidin 3(sin)triglucoside-5-glucoside ([M]^+^ ion at *m/z* 1141.32544) was also identified. The latter compound and cyanidin 3(caf)(fer)sophorosyl-5-glucoside, along with seven other acylated anthocyanidins (cyan-3(caf)(sin)soph-5-glu; cyan-3(*p*-coum)triglu-5-glu; cyan-3(*p*-coum)soph-5-(suc)glu; cyan-3(glucop-sin)(*p*-coum)soph-5-glu; cyan-3(glucop-sin)(fer)soph-5-glu; cyan-3(glucop-sin)(sin)soph-5-glu; cyan-3(fer)soph-5-(sin)glu)) were identified for the first time by Arapitsas et al. [64], who analyzed the red cabbage anthocyanin profile using HPLC/DAD-ESI/Qtrap MS.

Mizgier et al. [67] analyzed the phenolic compounds and antioxidant capacity of red cabbage extract, confirming that more than 80% was constituted from acylated compounds. The total anthocyanin content of the red cabbage extracts was reported to be equal to 175.1 mg/g dry weight, which was expressed as cyanidin-3-glucoside equivalents.

Wiczkowski et al. [63] studied the red cabbage anthocyanins profile and analyzed the correlation of antioxidant activity and acylated compounds, demonstrating that cyanidin 3-diglucoside-5-glucoside diacylated with sinapic acid had the highest radical-scavenging activities. Acylated cyanidin glycosides showed higher antioxidant capacity than the non-acylated form of cyanidin glycosides. In this case, the total content of anthocyanins in red cabbage was 2.32 mg/g on dry weight, which was calculated based on cyanidin equivalents [63]. Dominant forms of anthocyanins in this red cabbage were non-acylated compounds, which comprised 27.6% of the total red cabbage anthocyanins, whereas mono-acylated and diacylated anthocyanins covered 38.4% and 34.1%, respectively.

Other reports evidenced that red cabbage is an excellent vegetable containing a high content of anthocyanins, which were in the range from 40 to 188 mg Cy 3-glcE/100 g fresh weight [68,69].

Voća et al. [70], who analyzed the difference in chemical composition between cabbage cultivars, reported the highest content of anthocyanins in red cabbage extract (until 750.71 mg/kg fresh weight). In red cabbage cultivars, even 3.9 times higher antioxidant capacity was reported compared to the other cultivars. Another report estimated the total anthocyanins content of the red cabbage extract equal to 4984.13 ± 101.62 μg/g dry weight [71]. In addition, the flavonoids composition was analyzed by Voća et al. as well [70]; red cabbage contains, besides anthocyanins, other phenolic compounds also. In particular, the most abundant part of the extract (73.7%) consisted in C_6_C_3_ phenolic acids as sinapic acid (6325.025 ± 3.568 μg/g dry weight), ferulic acid (2768.48 ± 29.18 μg/g dry weight), and *p*-coumaric acid (4518.52 ± 15.83 μg/g dry weight), beyond C_6_C_1_ phenolic acids such as vanillic acid (5961.13 ± 29.08 μg/g dry weight) and protocatechuic acid (2881.17 ± 15.59 μg/g dry weight). Thus, sinapic acid covered 37.5% of phenolic acids.

Finally, non-anthocyanin flavonoids were found in a negligible portion, as they constituted only 0.38% of the extract. In particular, among flavonols, kaempferol appeared to mostly contain 17.371 ± 0.23 μg/g dry weight, followed by rutin and its aglycone quercetin, whose relative contents were estimated to be equal to 15.302 ± 0.96 and 4.359 ± 0.33 μg/g dry weight. The flavanol epigallocatechin constituted 23.965 ± 0.16 μg per g of dry extract.

## 4. Conclusions

Red cabbage is a rich source of phenolic acids and flavonoids. Among the latter, anthocyanins represent the most abundant class. Indeed, the full exploitation of the beneficial antioxidant efficacy of these compounds requires their extraction optimization. In this context, extractive procedures on red cabbage provided an extract rich in cyanidin-3-diglucoside-5-glucoside and its acylated compounds. In particular, UHPLC-Q-Orbitrap HRMS analysis highlighted the diversity in sinapoyl derivatives, and sinapic acid was also the most representative phenolic acid in the extract. The chemically profiled extract was screened for its antiradical and reducing capabilities, whereas its stability and bioaccesibility were proved to be preserved in an acid-resistant capsule. Indeed, based on the data acquired, improvements for all the evaluated bioactivities (TPC and antioxidant activity) were observed during the Pronase E and/or Viscozyme L phases of the in vitro gastrointestinal digestion of the red cabbage extract in an acid-resistant capsule. An increase in the colonic phase in TPC was also observed, as well as a similar trend for the other evaluated antioxidant activities. During gastrointestinal digestion, bioactive compounds could be metabolized by human colonic microflora, generating metabolites with greater bioactivity and more beneficial effects. Thus, the use of an acid-resistant capsule in the digestion process could protect bioactive compounds, ameliorating their bioaccessibility. In this context, red cabbage extract encapsulated in an acid-resistant capsule could be a valid alternative to produce a new nutraceutical formulation useful for preventing or slowing down oxidative stress-related diseases onset.

## Figures and Tables

**Figure 1 antioxidants-09-00955-f001:**
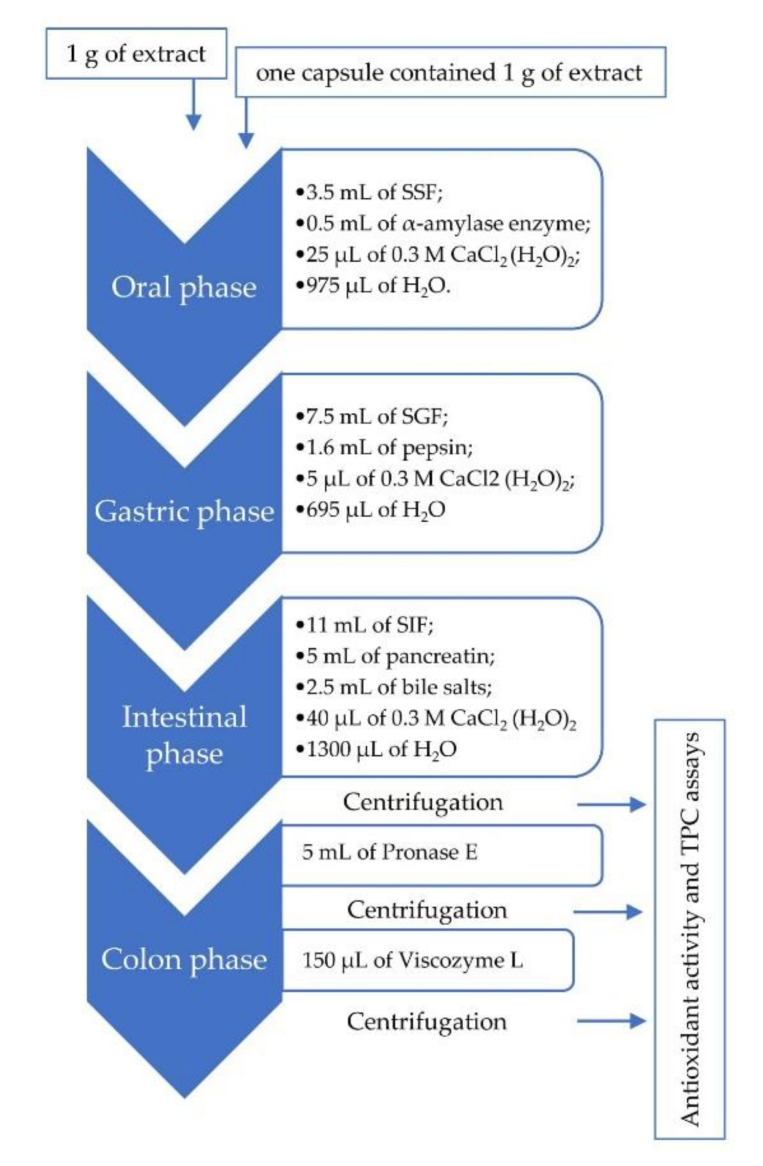
Overview and flow diagram of a simulated in vitro digestion method. SSF: simulated salivary fluid; SGF: simulated gastric fluid; SIF: simulated intestinal fluid.

**Figure 2 antioxidants-09-00955-f002:**
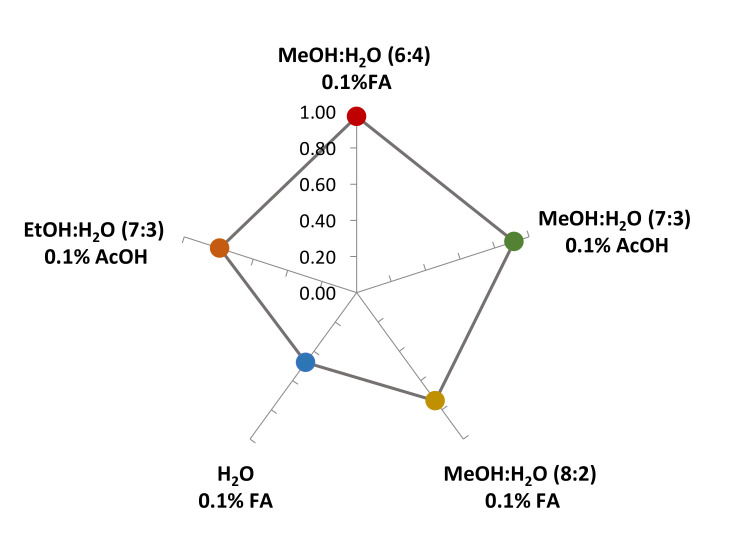
Relative Antioxidant Capacity Index (RACI) was used to integrate the antioxidant capacity values generated from the different applied methods.

**Table 1 antioxidants-09-00955-t001:** Total polyphenol content and antioxidant data of red cabbage extracts obtained using five different extractants. Values are reported as mean ± SD of independent experiments performed in triplicate. Statistic significance was calculated with two-way ANOVA analysis.

Red Cabbage Extract	TPC * (mg GAE/g)	ABTS *	DPPH * (mmol Trolox/kg)	FRAP *
MeOH:H_2_O (6:4) 0.1% FA	19.986 ± 0.132 ^a^	50.849 ± 2.955 ^d^	36.242 ± 0.068 ^f^	87.095 ± 0.699 ^h^
MeOH:H_2_O (7:3) 0.1% AcOH	17.591 ± 0.721 ^b^	49.978 ± 1.408 ^d^	34.466 ± 0.168 ^f^	85.379 ± 0.349 ^h^
MeOH:H_2_O (8:2) 0.1% FA	17.541 ± 0.304 ^b^	45.128 ± 1.065 ^e^	24.003 ± 0.056 ^g^	68.759 ± 0.349 ^i^
H_2_O 0.1% FA	15.798 ± 0.566 ^c^	47.906 ± 1.411 ^d^	23.498 ± 0.345 ^g^	67.833 ± 1.325 ^i^
EtOH:H_2_O (7:3) 0.1% AcOH	16.262 ± 0.528 ^c^	49.324 ± 0.761 ^d^	24.701 ± 0.166 ^g^	81.884 ± 1.277 ^h^

* results are referred to dry weight (dw). ^a–h^ Mean values with different superscript letters are significantly. different by Tukey–Kramer multiple comparison test.

**Table 2 antioxidants-09-00955-t002:** Recent surveys reporting the total phenolic content (mg GAE/g) in red cabbage extract. GAE: gallic acid equivalents.

References	TPC (mg GAE/g) dw *	Extraction Solvent	Sample Origin
Leja et al., 2010 [30]	3.90–31.11	80% MeOH	Poland
Upadhyay et al., 2016 [31]	28.25	80% MeOH	India
Fusari et al., 2020 [32]	0.099	Ultrapure H_2_O	Argentina
Erken et al., 2017 [33]	313.73	EtOH/acetone	Turkey
Murador et al., 2016 [27]	3.56	0.5% HCl in MeOH	Brazil
Podsędek et al., 2017 [34]	10.1–19.6	70% MeOH	Poland
Podsędek et al., 2006 [35]	16.8–21.42	70% MeOH	Poland
Oroian et al., 2017 [36]	84.75	MeOH	Romania
Tanongkankit et al., 2010 [37]	496.92–739.24	acetone–H_2_O (1:1, *v/v*)	Thailand
Jaiswal et al., 2012 [38]	18.45	70% MeOH	Ireland
Kusznierewicz et al., 2007 [39]	2.4–4.9	0.1% HCl (1 N) in MeOH	Poland/Belgium Germany/England
Tabart et al., 2018 [28]	18.51	acetone/H_2_O/AcOH, (70:28:2, *v/v/v*)	Belgium
Cruz et al., 2016 [40]	89.33–116	70% MeOH/ H_2_O boiled	Brazil
Caramês et al., 2020 [41]	48.37–87.12	MeOH/H_2_O/AcOH (0.58:0.38:0.04, *v/v/v*)	Brazil

* results are adjusted and expressed in the same measurement unit as mg GAE/g dry weight.

**Table 3 antioxidants-09-00955-t003:** Total polyphenol content of red cabbage extract compared to that of the same extract encapsulated in an acid-resistant capsule. Values are reported as mean ± SD of independent experiments performed in triplicate. Statistic significance was calculated with two-way ANOVA analysis.

Red Cabbage In Vitro GI Digestion		TPC (mg GAE/g)	
Phase		Extract	Capsule *
**1**	Intestinal phase	22.287 ± 0.295 ^a^	22.738 ± 0.339 ^a^
**2**	Pronase E	0.124 ± 0.003 ^b^	4.434 ± 0.069 ^c^
**3**	Viscozyme L	0.994 ± 0.060 ^d^	0.102 ± 0.022 ^e^

* results are referred to dry weight (dw) of extract; red cabbage extract was encapsulated in polyethylene capsule and used in in vitro digestion process. ^a–e^ Mean values with different superscript letters are significantly different by Tukey–Kramer multiple comparison test.

**Table 4 antioxidants-09-00955-t004:** Antioxidant data by means of 2,2-azinobis (3-ethylbenzothiazoline-6-sulphonic acid) diammonium salt (ABTS), 1,1-diphenyl-2-picrylhydrazyl (DPPH), and ferric reducing/antioxidant power (FRAP) methods of red cabbage extract compared to that of the same extract encapsulated in an acid-resistant capsule. Values are reported as mean ± SD of independent experiments performed in triplicate.

		Red Cabbage In Vitro GI Phase		
		1 Intestinal phase	2 Pronase E	3 Viscozyme L
**ABTS** **(mmol Trolox/kg)**	**Capsule**	76.755 ± 1.483 ^a^	0.682 ± 0.044 ^b^	2.600 ± 0.220 ^d^
	**Extract**	78.513 ± 1.783 ^a^	12.820 ± 0.949 ^c^	2.626 ± 0.067 ^d^
**DPPH** **(mmol Trolox/kg)**	**Capsule**	44.985 ± 2.547 ^e^	0.150 ± 0.003 ^f^	4.074 ± 0.126 ^h^
	**Extract**	45.762 ± 1.773 ^e^	7.268 ± 1.095 ^g^	0.008 ± 0.001 ^i^
**FRAP** **(mmol Trolox/kg)**	**Capsule**	90.778 ± 2.128 ^j^	2.055 ± 0.355 ^k^	28.954 ± 1.793 ^m^
	**Extract**	91.958 ± 1.502 ^j^	132.931 ± 0.939 ^l^	0.528 ± 0.074 ^b^

* Results are referred to dry weight (dw) of extract; ^a^^–m^ Mean values with different superscript letters are significantly different by Tukey–Kramer multiple comparison test.

**Table 5 antioxidants-09-00955-t005:** Quantitation of the main anthocyanin compounds in the investigated red cabbage extract, as well as of the most abundant non-anthocyanin flavonoids and phenolic acids. Results are expressed as mean ± SD from three independent determinations.

Compound	Red Cabbage Content (mg/kg)	SD
**Anthocyanins**		
Cyanidin 3-diglucoside-5-glucoside	2344.684	9.198
Cyanidin 3-soph-5-xyloside	24.660	5.115
Cyanidin 3,5-diglucoside	306.750	12.450
Cyanidin 3-galactoside	7.000	1.170
Cyanidin 3-(sin)soph-5-glucoside	514.153	11.001
Cyanidin 3-(sin)triglucoside-5-glucoside	131.554	4.225
Cyanidin 3-(glucofer)-diglucoside-5-glucoside	10.549	0.617
Cyanidin 3-(*p*-coum)-glucoside-5-glucoside	13.395	3.542
Cyanidin 3-(caf)-diglucoside-5-glucoside	22.115	2.892
Cyanidin 3-(fer)-glucoside-5-glucoside	28.009	2.759
Cyanidin 3-(sin)glucoside-5-glucoside	153.803	11.370
Cyanidin 3-(*p*-coum)-diglucoside-5-glucoside	340.812	8.372
Cyanidin 3-(fer)soph-5-glucoside	268.096	9.925
Cyanidin	112.793	2.745
Cyanidin 3-(sin)-diglucoside-5-glucoside	727.842	13.723
Cyanidin 3-(caf)(*p*-coum)-diglucoside-5-glucoside	5.680	0.056
Cyanidin 3-(caf)(sin)soph-5-glucoside	31.820	8.544
Cyanidin 3-(sin)(*p*-coum)soph-5-glucoside	186.841	11.523
Cyanidin 3-(sin)(fer)soph-5-glucoside	195.990	11.289
Cyanidin 3-(sin)(sin)soph-5-glucoside	506.567	8.600
**Other flavonoids**		
Epicatechin	1.544	0.232
Epigallocatechin	23.965	0.161
Rutin	15.302	0.963
Kaempferol	17.371	0.228
Quercetin	4.359	0.326
Genistein	24.122	1.471
**Phenolic acids**		
Caffeic acid	39.356	1.250
Chlorogenic acid	55.316	5.581
*p-*Coumaric acid	3689.235	2.356
Ferulic acid	2533.965	3.161
Protocatechuic acid	2056.230	2.102
Sinapic acid	6325.025	3.568
Syringic acid	17.069	0.532
Vanillic acid	2136.250	3.256

* Results are referred to dry weight (dw) of extract; sin = sinapoyl; soph = sophoroside; fer = feruloyl; *p*-coum = *p*-coumaroyl; caf = caffeoyl; glucofer = glucoferoyl.

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
