# Peer review of "Colon Bioaccessibility under In Vitro Gastrointestinal Digestion of a Red Cabbage Extract Chemically Profiled through UHPLC-Q-Orbitrap HRMS"

_antioxidants, 2020, doi:10.3390/antiox9100955_

Round 1
Reviewer 1 Report
In this manuscript, the authors present the total phenolic content and the antioxidant activity of red cabbage extracts, as well as a quantitative profile analysis of the bioactive compounds, and the effect of simulated gastrointestinal digestion on the total phenolic content. The extract was used directly or encapsulated in polyethylene capsules.
The work presented may be of relevance if complemented with additional information and rewritten for clarity. Specific comments are detailed below by the order of appearance in the manuscript:
1 - Introduction like test is present outside the introduction, making the distinction between original and data from literature a difficult task. This is particularly important for the results and discussion section, that includes introduction, results and discussion. The results section must be focused on original data and their brief interpretation. A separated discussion is encouraged with a comparison of the results obtained with literature data together with an overall critical evaluation. All information regarding general aspects (for example lines 247 to 268) must be moved to the introduction.
2 - The description of the method followed for the simulated Gastrointestinal Digestion may be improved. Given the complexity of the procedure, it would be better presented in an workflow, with a clear identification of which samples are being characterized.
3 - Paragraph in lines 269-270 is not clear and is out of place. Do the authors mean that all the results were divided by the mass of extract used in the specific assay to normalized to the same amount of extract? This information is somewhat redundant because the results are already given /g or /kg. For a better clarification, the * on the tables should indicate that it refers to dry weight of extract. What is also relevant is to specify if the replicates are independent (different extracts) or not.
4 - The method used to calculate the Relative Antioxidant Capacity Index shown in Figure 1 should be explained, and/or the relevant references should be given. The expression "a gradually increasing decrease" must be replaced by a simpler one. This result should be interpreted, why does methanol fractions above 60% lead to extracts with smaller antioxidant capacity? What is the effect of using FA or AcOH?
5 - Table 2. Most of the information presented in this table is not relevant for the interpretation of the results shown. The authors may wish to present an overview of the methods used previously, maybe better in the introduction. But this information is only relevant if it is interpreted, namely for the identification of the conditions to follow in this work. The results obtained in this work should then be compared with data from literature and eventual differences discussed.
6 - Section 3.2. This section must be significantly changed to improve clarity.
For example, the sentence "Results showed an increase in the bioaccessibility of bioactive compounds after the colonic stage" is very difficult to understand. How do the authors access bioaccessibility? Is it the level of TPC in the supernatant at the different stages in the digestion? This level of TPC in the supernatant after action of the enzymes mimicking the colonic stage is much smaller than after the intestinal phase...
How can the TPC after the intestinal phase be larger than that of the extract? Was the extract not solubilized when characterized? Or chemical modifications occurring during digestion lead to a higher TPC in GAE equivalents? How is the TPC measurement affected by the encapsulation of the extract in the capsule?
The authors should discuss the effectiveness of the "acid-resistant" capsule during the intestinal phase and in the presence of the enzymes mimicking the colonic stage.
The sentence "..., polyphenols present a poor bioavailability because of low water solubility." requires scientific support.
7 - section 3.3. It is not clear what was the sample characterized. Was it the extract? An extension of this analysis to each of the other samples (after intestinal digestion and treatment with the enzymes mimicking the colonic stage) would be of high relevance.
Author Response
Response to reviewers
Manuscript ID: antioxidants-933501
Type of manuscript: Article
Title: Colon bioaccessibility under in vitro gastrointestinal 3 digestion of a red cabbage extract chemically profiled 4 through UHPLC-Q-Orbitrap HRMS
Reviewer 1
Comments and Suggestions for Authors
In this manuscript, the authors present the total phenolic content and the antioxidant activity of red cabbage extracts, as well as a quantitative profile analysis of the bioactive compounds, and the effect of simulated gastrointestinal digestion on the total phenolic content. The extract was used directly or encapsulated in polyethylene capsules.
The work presented may be of relevance if complemented with additional information and rewritten for clarity. Specific comments are detailed below by the order of appearance in the manuscript:
1 - Introduction like test is present outside the introduction, making the distinction between original and data from literature a difficult task. This is particularly important for the results and discussion section, that includes introduction, results and discussion. The results section must be focused on original data and their brief interpretation. A separated discussion is encouraged with a comparison of the results obtained with literature data together with an overall critical evaluation. All information regarding general aspects (for example lines 247 to 268) must be moved to the introduction.
-As suggested by reviewer 1, the authors deleted results to the introduction section and move the lines 247-268 in the introduction part.
2 - The description of the method followed for the simulated Gastrointestinal Digestion may be improved. Given the complexity of the procedure, it would be better presented in a workflow, with a clear identification of which samples are being characterized.
-As suggested by reviewer 1, the authors improved the description of simulated gastrointestinal digestion and added the procedure in a workflow.
3 - Paragraph in lines 269-270 is not clear and is out of place. Do the authors mean that all the results were divided by the mass of extract used in the specific assay to normalized to the same amount of extract? This information is somewhat redundant because the results are already given /g or /kg. For a better clarification, the * on the tables should indicate that it refers to dry weight of extract. What is also relevant is to specify if the replicates are independent (different extracts) or not.
- As suggested by reviewer 1, the authors deleted the sentence and specified on the tables that the results were referred to the dry weight of the extract. Moreover, following the indications given by reviewer the authors specified that the replicates were independents.
4 - The method used to calculate the Relative Antioxidant Capacity Index shown in Figure 1 should be explained, and/or the relevant references should be given. The expression "a gradually increasing decrease" must be replaced by a simpler one. This result should be interpreted, why does methanol fractions above 60% lead to extracts with smaller antioxidant capacity? What is the effect of using FA or AcOH?
-As suggested by reviewer 1, the authors provided a relevant reference for the calculation of the Relative Antioxidant Capacity Index. The expression "a gradually increasing decrease" was replaced by a simpler one. Presumably, the mixture MeOH:H2O (6:4, v/v) results to be a good compromise applicable to the extraction of red cabbage bioactive compounds. By increasing the percentage of methanol, the TPC content decrease. This effect could be due to the major solubility of anthocyanins, contain in high quantity in red cabbage, in a high percentage of acidified water whereas polyphenols are more soluble in methanol.
5 - Table 2. Most of the information presented in this table is not relevant for the interpretation of the results shown. The authors may wish to present an overview of the methods used previously, maybe better in the introduction. But this information is only relevant if it is interpreted, namely for the identification of the conditions to follow in this work. The results obtained in this work should then be compared with data from literature and eventual differences discussed.
-As suggested by reviewer 1, the authors added some discussions in this section.
6 - Section 3.2. This section must be significantly changed to improve clarity.
-As required by reviewer 1, the authors clarify this section.
*For example, the sentence "Results showed an increase in the bioaccessibility of bioactive compounds after the colonic stage" is very difficult to understand. How do the authors access bioaccessibility? Is it the level of TPC in the supernatant at the different stages in the digestion? This level of TPC in the supernatant after action of the enzymes mimicking the colonic stage is much smaller than after the intestinal phase...
-As rightly suggested by reviewer 1, the authors clarify the sentence as:” An improvement of bioactivity was observed for the extract digested in an acid-resistant capsule after the colonic stage compared to the extract digested without capsule”. The bioaccessibility of total phenolic compounds was evaluated using the Folin-Ciocalteu method by measuring the TPC in the supernatant at the different stages of the digestion.
*How can the TPC after the intestinal phase be larger than that of the extract? Was the extract not solubilized when characterized? Or chemical modifications occurring during digestion lead to a higher TPC in GAE equivalents? How is the TPC measurement affected by the encapsulation of the extract in the capsule?
-Exactly, chemical modifications occurring during gastrointestinal digestion including the activity of gut microbiota, lead to a releasing of smaller compounds, the principal responsible for the increased antioxidant activity. Although polyphenols were poorly absorbed in the duodenum, they can exert their antioxidant activities in the lower gut which is able to metabolize these compounds, generating metabolites with greater activity. Our hypothesis to explain increases of polyphenols in the colon stage is that after the in vitro GI digestion, polyphenols might be released from glucose residues by the activities of Pronase and Viscozyme, showing the increase of free polyphenols observed by Folin-Ciocalteu.
-Moreover, the use of an acid-resistant capsule in the digestion process allowed us to avoid the consequence of gastric ambient on the bioactive molecules and act as a protective agent. The acid-resistant capsules represent a useful strategy to conduct bioactive molecules until the intestine district, where the bioactive compounds are favorably absorbed, and exert their activity.
*The authors should discuss the effectiveness of the "acid-resistant" capsule during the intestinal phase and in the presence of the enzymes mimicking the colonic stage.
-As required by reviewer 1, the authors added further details for better understanding.
*The sentence "..., polyphenols present a poor bioavailability because of low water solubility." requires scientific support.
-As required by reviewer 1, the authors added an opportune reference.
7 - section 3.3. It is not clear what was the sample characterized. Was it the extract? An extension of this analysis to each of the other samples (after intestinal digestion and treatment with the enzymes mimicking the colonic stage) would be of high relevance.
- In the current study, the sample characterized refers to MeOH:H2O (6:4, v/v) extract. To clarify this, the authors add this missing information in the manuscript. An extension of this analysis to each of the other samples is considered an interesting approach for future work.
The authors thank reviewer 1 for evaluating our manuscript.
Reviewer 2 Report
I have carefully read the manuscript entitled “Colon bioaccessibility under in vitro gastrointestinal digestion of a red cabbage extract chemically profiled through UHPLC-Q-Orbitrap HRMS” by Izzo et al, (Manuscript ID: antioxidants-933501). My comments are as follows:
Many foods typical of the Mediterranean diet contain significant amounts of anthocyanins, which are thought to contribute to the inverse relationship between fruit and vegetable intake and incidence of chronic diseases. Although the underlying mechanism(s) of their protection require further elucidation, it has been proposed that their beneficial health effects are related to their antioxidant potential.
This stydy, aims to evaluate and to compare the antioxidant capacity and the total polyphenol content of i) a red cabbage extract ii) the same extract encapsulated in an acid-resistant capsule. Subsequently, an in vitro gastrointestinal digestion system was utilized to follow the extracts’ metabolism in conditions simulating the phases of digestion.
In my opinion, the experiments of this study are well-designed and well-conducted, and the results are quite interesting. However, the authors’ proposal for the pharmaceutical exploitation of the above extracts, is somewhat an exaggeration, considering that this is an in vitro conducted study. Nevertheless, if the authors wish to support this perspective, in my opinion, they should explain in more detail:
- Is there a direct relationship between the antioxidant capacity (as estimated in vitro) and the cytoprotective effect of a compound?
- What are the key properties a compound should exhibit, in order to have cytoprotective actions?
- Are there studies showing that representative agents of the above extracts meet these criteria?
- Finally, as it the authors mention in their manuscript that red cabbage extracts may counteract oxidative stress and protect against oxidative stress-related diseases, a more detailed definition of the concept of oxidative stress should be presented.
Author Response
Response to reviewers
Manuscript ID: antioxidants-933501
Type of manuscript: Article
Title: Colon bioaccessibility under in vitro gastrointestinal 3 digestion of a red cabbage extract chemically profiled 4 through UHPLC-Q-Orbitrap HRMS
Reviewer 2
Comments and Suggestions for Authors
I have carefully read the manuscript entitled “Colon bioaccessibility under in vitro gastrointestinal digestion of a red cabbage extract chemically profiled through UHPLC-Q-Orbitrap HRMS” by Izzo et al, (Manuscript ID: antioxidants-933501). My comments are as follows:
Many foods typical of the Mediterranean diet contain significant amounts of anthocyanins, which are thought to contribute to the inverse relationship between fruit and vegetable intake and incidence of chronic diseases. Although the underlying mechanism(s) of their protection require further elucidation, it has been proposed that their beneficial health effects are related to their antioxidant potential.
This stydy, aims to evaluate and to compare the antioxidant capacity and the total polyphenol content of i) a red cabbage extract ii) the same extract encapsulated in an acid-resistant capsule. Subsequently, an in vitro gastrointestinal digestion system was utilized to follow the extracts’ metabolism in conditions simulating the phases of digestion.
In my opinion, the experiments of this study are well-designed and well-conducted, and the results are quite interesting. However, the authors’ proposal for the pharmaceutical exploitation of the above extracts, is somewhat an exaggeration, considering that this is an in vitro conducted study. Nevertheless, if the authors wish to support this perspective, in my opinion, they should explain in more detail:
- Is there a direct relationship between the antioxidant capacity (as estimated in vitro) and the cytoprotective effect of a compound?
- What are the key properties a compound should exhibit, in order to have cytoprotective actions?
- Are there studies showing that representative agents of the above extracts meet these criteria?
- Finally, as it the authors mention in their manuscript that red cabbage extracts may counteract oxidative stress and protect against oxidative stress-related diseases, a more detailed definition of the concept of oxidative stress should be presented.
- As suggested by reviewer 2, the authors added in the text details about cytoprotective effect of Anthocyanins. “According to Sies et al., [25] the oxidative stress represents “a disturbance in the pro-oxidant–antioxidant balance in favor of the former, leading to potential damage". Anthocyanins are known to have a wide range of health‐promoting properties for human health including cytoprotective activity, which might be due to the ability of anthocyanins to decreasing cell death, LDH release, caspase 3 activation, and DNA damages [26]. Moreover, scientific reports support an increase in the cytoprotective effect as a result of the anthocyanin-rich diet [27].”
The authors thank reviewer 2 for evaluating our manuscript.
Reviewer 3 Report
Dear Authors,
I marked some corrections and wrote some comments to improve the document in the pdf of the manuscript.
It will be sent to Authors by the Editor.
Best regards

Author Response
Response to reviewers
Manuscript ID: antioxidants-933501
Type of manuscript: Article
Title: Colon bioaccessibility under in vitro gastrointestinal digestion of a red cabbage extract chemically profiled 4 through UHPLC-Q-Orbitrap HRMS
Reviewer 3
Comments and Suggestions for Authors
Dear Authors,
I marked some corrections and wrote some comments to improve the document in the pdf of the manuscript.
It will be sent to Authors by the Editor.
Best regards
-The authors modified the manuscript in accordance with reviewer 3 suggestions.
The authors thank reviewer 3 for evaluating our manuscript.
Round 2
Reviewer 1 Report
The authors have addressed adequately the concerns raised by the reviewer.
This manuscript is a resubmission of an earlier submission. The following is a list of the peer review reports and author responses from that submission.